# Can Social Media Profiles Be a Reliable Source of Information on Nutrition and Dietetics?

**DOI:** 10.3390/healthcare10020397

**Published:** 2022-02-20

**Authors:** Paweł Kabata, Dorota Winniczuk-Kabata, Piotr Maciej Kabata, Janusz Jaśkiewicz, Karol Połom

**Affiliations:** 1Department of Surgical Oncology, Faculty of Medicine, Medical University of Gdańsk, 80-214 Gdańsk, Poland; januszj@gumed.edu.pl (J.J.); surgoncolclub@gmail.com (K.P.); 2Department of Anesthesiology and Intensive Care, Faculty of Medicine, Medical University of Gdańsk, 80-214 Gdańsk, Poland; dorota.winniczuk@gmail.com; 3Pomeranian Centre of Toxicology, 80-104 Gdańsk, Poland; kabasny@gmail.com

**Keywords:** social media, instagram, nutrition, education

## Abstract

Background: Social media are growing worldwide platforms for unlimited exchange of various content. Owing to their accessibility and short form, they can be utilized as usable, wide-range communication and information tools for companies, scientific communities, patient advocacy organizations, and special interest groups. This study aimed to investigate whether Instagram^®^ profiles can be reliable sources of information and knowledge about nutrition and dietetics. Materials and Methods: Random identification of nutrition-related posts was performed using a built-in website search engine. Posts were searched by five popular hashtags: #nutrition, #nutritionist, #instadiet, #diet, and #dietitian, 250 newest posts of each. Advertisement posts were discarded. Each eligible post was then categorized (dietetics, fitness, motivation, other) and assessed with regard to the quality of nutrition information provided (five levels from none to good quality), popularity (number of followers, likes, and comments), and engagement measures (like, comment, and engagement ratio). Results: A total of 1189 posts were reviewed. The overall quality of the content regarding nutritional knowledge was extremely low (93.9% of all posts), also when divided into categories. Among all posts, 63.8% were categorized as “nutrition and dietetics”, while “fitness”, “motivation”, and “other” categories comprised 8.2%, 4.8%, and 23.2% of the posts, respectively. Posts recognized as dietetics were the most liked (mean *n* = 116 likes per post) and of the highest quality. However, those motivational raised the greatest degree of engagement (32.7%). Posts with cooking recipes were the most commented. Conclusions: Random post search cannot provide viewers with valuable nutrition information. A dedicated search for high-quality professional profiles is preferred to obtain quality information.

## 1. Introduction

Social media are fast-growing worldwide platforms connecting millions of people and enabling the rapid exchange of different content. As they are easily accessible, short in form, and very user-friendly, they can be utilized for various purposes, depending on the concept standing behind them. Almost 4.5 billion people worldwide use them daily to update their friends and acquaintances with important circumstances occurring in their lives, share visual, video, or musical content, and follow well-known people. Social media such as Twitter^®^ can also be used for professional and scientific purposes, enabling the fast spread of novel scientific findings, research results, and political debates. This use inevitably led to live-tweeting during large scientific congresses, allowing the dissemination of knowledge to vast groups of recipients. [1]

These platforms offer instant, 24/7 access to large groups of people, and are also great marketing machines used widely by companies and manufacturers of all branches of various industries. Nevertheless, apart from their use for marketing purposes, they have also been adopted for health promotion by scientific societies, patient advocacy organizations, and other medical and paramedical professionals. Many of these accounts are led by professionals willing to promote and spread well-documented, evidence-based medical and paramedical knowledge put in a more people-friendly, less scientific form. This approach resulted in leading medical influencer accounts reaching tens to hundreds of thousands of followers in almost all fields of medicine.

Nutrition and dietetics seem to be medical and paramedical knowledge fields that could benefit from social media. However, with the massive popularity of the so-called fit lifestyle and healthy eating promotion, their reach is far beyond medical knowledge and crosses with sports and bodybuilding communities. Therefore, platforms such as Instagram^®^, which was initially designed to share photos, enabling their editing, and applying enhancement filters, appear to be perfect for this purpose. Abundant colorful photos of meals and attractive bodies of fitness and bodybuilding promoters act as teasers to explore the content and eventually follow their creator’s profile. However, this popularity might also create information chaos and difficulties differentiating good-quality content from poor ones, all grouped under specific hashtags. 

A hashtag is a special type of identifier used in social media. It is a keyword or a phrase used to describe the content followed by a hash (#) sign [2]. Using hashtags enables social media users to identify the content of their interest and enables platform algorithms to make user-oriented offers, commercials, or recommend popular accounts to be followed. In 2017, Brady et al. showed that using a specific *#colorectalsurgery* hashtag for an online campaign can rapidly create a worldwide community of specialists in a particular field of interest, helping to spread the knowledge and bridge the barriers [3]. Its usability to search for medical information in social media channels, such as breast cancer, has also been studied by other authors [4]

This study aimed to determine whether random hashtag search on Instagram^®^ for nutrition and dietetics content can provide the user with good-quality data and information on these subjects and correlate with posts’ popularity and performance. 

## 2. Materials and Methods

In this cross-sectional study, five popular nutrition-related hashtags—#nutrition, #nutritionist, #instadiet, #diet, and #dietitian—were identified using the Instagram^®^ webpage search machine (www.instagram.com, accessed on 11 January 2022). At the time of the search, these hashtags were used over 45.5 million, 2.94 million, 897 thousand, 63.7 million, and 1.56 million times, respectively. 

Instagram^®^’s (Meta Platforms Ireland Ltd, Dublin, Ireland) search engine enables its user to search for content in one of two ways —most recent posts or most popular ones. The search engine has been set to time-dependent search, which is not influenced by the popularity of a post and, therefore, free from any promoting actions by the website’s algorithm, to provide random post identification. The search in this study occurred on 25–26 November 2018. Within 24 h, 250 newest posts from each hashtag were recorded and evaluated. Initial evaluation included the eligibility of each post for further assessment. Posts containing advertisements and commercial offers, abusive or potentially hazardous content, and multiplied using more than one of the searched hashtags were discarded (*n* = 61). Eventually, 1189 eligible posts were assessed again but not sooner than 48 h after the initial search, to enable the adequate performance of posts and viewers’ reactions to occur. 

Subsequently, the research team read and categorized each post, depending on its content. Categories were not preestablished and were created in real time during the search, based on the type of content found in the posts. The following categories were established: “nutrition and dietetics”; “fitness”; “motivation”; “other.” Posts qualified as “other” contained any other content unsuitable for one of the above categories. Having found that each of these major categories could cover a broader range of various presentations, we decided to subcategorize them to describe the content of each post as precisely as possible. As a result, the following seven areas of interest were developed: “dietary advice”; “general advice and coaching”; “cooking recipe”; “standalone photo without textual content”; “motivational quotes”; “fitness”; “other.” 

The quality of the content and its educational value regarding nutritional and dietary knowledge were assessed. Due to a lack of dedicated tools to evaluate the quality of this type of content, a 5-point Likert scale was adopted for this purpose, and the quality was graded as none, very low, low, moderate, and good. This scale is commonly used in questionnaires to express one’s attitude towards a question or a statement. It assumes that the strength of an attitude is linear and can be measured. In this study, we aimed to answer the question “What is the quality of the post?” When assessing the post quality, the amount of nutrition-related information and its quality concerning current knowledge were considered. For example, if a post contained only a photograph of food without any informative content, and the “nutrition” hashtag was added, such post was qualified as “none” regarding its educational value. On the other hand, posts containing well-prepared, reliable, and verifiable information were treated as good-quality ones.

We decided to assess the performance of each post independently of quality assessment. Basic performance measures of each post included the number of likes and comments and the number of account followers at the time of assessment. Due to considerable differences in the number of accounts’ followers, the study group was also divided into follower groups for further analyses < 1000, 1–5 thousand, >5–10 thousand, >10–50 thousand, and >50 thousand. In addition, the following advanced performance measures were also calculated: the like-to-follower ratio (LFR), which is used to reflect the percentage of account followers who react by “liking” the post; comment-to-like (CLR) ratio, which reflects the percentage of followers likely to comment on the liked content; overall engagement rate (ER), counted as the sum of likes and comments divided by the number of followers. These measures are commonly used by popular social media management tools to assess performance and are available for professional accounts. 

Multidirectional analyses were performed (Figure 1). All posts grouped in major categories and subcategories regardless of their hashtag identification were analyzed separately. In addition, independent analyses were performed for posts grouped under each hashtag.

### 2.1. Ethical Considerations

A local university Ethical Committee (Institutional Review Board) was contacted for ethical review of the study. However, as the study did not include human participants, and all of the gathered data were derived from a publicly available website, the study did not require ethical committee approval. 

### 2.2. Statistical Analysis

Statistical analyses were performed with SPSS 22.0 software (SPSS Inc, Chicago, IL, USA). The normality of the data was calculated using the Shapiro–Wilk test. All numerical measures were presented as exact values. Data were analyzed for the whole group, as well as for each hashtag group. The frequency of each hashtag appearance within a category or subcategory was presented as a percentage. Means were calculated and presented. One-way ANOVA analysis was performed to compare performance measures between categories and subcategories and follower and hashtag groups. Spearman’s rank correlation test was performed to determine whether the quality of the content affected the performance measures of the posts. *p*-value < 0.05 was considered significant. 

## 3. Results

### 3.1. Major and Subcategory Analysis

A total of *n* = 1189 posts were reviewed. Those qualified as “nutrition and dietetics” were the most represented ones (*n* = 759; 63.8%), whereas “fitness”, “motivation”, and “other” accounted for 8.2%, 4.8%, and 23.2%, respectively. When subcategorized, standalone photos were the most common (*n* = 509; 42.8%), followed by “other” (22.7%), “dietary advice” (15.6%), “cooking recipe” (10.0%), “motivation” (5.6%), “coaching” (3.0%), and “fitness” (0.3%). 

The same trend was observed when major categories were divided and grouped under each hashtag. Therefore, “nutrition and dietetics” accounted for 47.0% of posts marked with #nutrition, up to 79.8% of #instadiet posts. The appearance of subcategories within hashtag groups was also similar. Thus, 54.9% of #nutrition, 53.8% of #instadiet, 51.3% of #diet, and 33.2% of #dietitian posts included only a photo without any textual content (Table 1).

### 3.2. Content Quality Analysis

The overall quality of the content was poor, with 70.4% (*n* = 838) of all posts qualified as none or very low quality and 23.5% (*n* = 280) as low. Only 5.8% (*n* = 69) were of moderate and <1% (*n* = 1) of good quality. Assessment within category groups showed that “nutrition and dietetics” posts were slightly higher but still very low quality in major and subcategories. Although differences in quality between categories were significant (*p* < 0.001), the same trend was not observed between hashtag groups. The quality positively correlated with the number of likes and comments under each post (*p* < 0.001), as well as the number of account followers (*p* = 0.026); however, the strength of the correlation was not high. At the same time, this was not observed when correlation was assessed for follower groups. No significant correlation between quality and other measures—LFR, CLR, or ER—was observed (Table 2). 

### 3.3. Performance Analysis of Posts

The mean number of accounts’ followers, likes, and comments in the whole group was *n* = 3005 (range: 1–221941), *n* = 110 (range: 3–8549), and *n* = 5 (range: 0–196), respectively. In the hashtag groups, the mean number of followers varied from *n* = 1070 (#instadiet) to *n* = 4078 (#nutritionist), likes from *n* = 38 (#instadiet) to *n* = 162 (#nutrition), and comments from *n* = 2 (#instadiet) to *n* = 8 (#dietitian). 

Analysis of performance measures of post categories within major and subcategories revealed that those qualified as “motivation” raised the highest viewers’ attention in both. Among major categories, dietetic posts were more liked (*p* = NS), and motivational posts were the ones with the highest LFR (30.6%) and ER (32.7%) values, and these differences were statistically significant (*p* = 0.03). Additionally, among subcategories, apart from the highest LFR and ER (*p* < 0.001), these posts also gained the highest number of likes (*p* < 0.001). At the same time, posts containing cooking recipes were the most commented, with the highest number of comments (*p* < 0.001) and comment ratio values (*p* = 0.003). Detailed results of performance measures for major categories can be found in Table 3 and for subcategories in Table 4. 

## 4. Discussion

Social media are rapidly growing, easily accessible, 24/7 worldwide internet platforms providing instant access to vast numbers of people. Though initially aimed for social interacting, chatting, and exchanging ideas and data [5], they are currently more often used for other purposes such as marketing, advertising, business solutions, education, dissemination of knowledge. Moreover, with many different types of content—from short Twitter^®^ communications to full-length videos—they enable the viewers to decide which type of content suits them best [6]. 

Currently, Twitter is the most used social medium among certain groups of medical professionals [7]. However, there is a need for a nonprofessional medium able to raise awareness by passing the knowledge to medicine-naïve people. This need has been shown by Vander Wyst et al. in a study in which Facebook^®^ was utilized to spread knowledge on proper nutrition among pregnant, low-income adolescents, and adults [8]. Furthermore, in a study assessing content posted by cancer survivors on Twitter^®^ and Instagram^®^, Cherian et al. found that a more personal and narrative approach dominated Instagram. In contrast, Twitter^®^ posts concentrated on more factual aspects [9]. Therefore, we decided to study Instagram^®^, which is the fifth-most popular social medium after Facebook^®^, Youtube and two communicators (WhatsApp^®^ and Messenger^®^) [10] and offers more user-friendly content. Furthermore, the photography-aimed specification of Instagram^®^ seems to correspond with visually attractive images of foods that can build proper attention toward the content [11]. 

In this study, by searching only five popular hashtags, we received a joint response of over 114 million posts regarding the searched topic, making it impossible to sort through them. As initially expected, the quality of the posts revealed in this random search, stratified by the time of publication, was very poor. In our opinion, this was caused by the very high number of posts published using these popular identifiers. Most of these were not meant for educational purposes but followed typical social media content-sharing practices. Therefore, those supposed to bring educational value for their viewers are lost in the significant number of competing posts. Although the fast growth of social media in recent years extorted improvements in the quality of the content to attract viewers and enable accounts’ expansion, there is still a considerable number of poor-quality data, which may cause informational chaos without bringing any additional value. In our study, this was visible in the positive correlation of account size with the quality of the post, followed by a higher number of likes and comments. However, it is not visual features alone that decide what is looked for but also educational content. A study by Alassiri assessing social media search for medical information in the population of Saudi Arabia revealed that, among specialties, nutrition and a healthy lifestyle were the most searched features. 

Interestingly, although most study respondents expressed a positive attitude toward this form of education, over 73% of them questioned its credibility [12]. Moreover, such educational content is available in social media, also Instagram^®^, but a random search, as performed in our study, hardly allows the user to access it. To find helpful quality information on nutrition, one has to manually search the app or website for dedicated profiles. 

Nowadays, nutrition and dietetics, apart from being included in clinical medicine, are inevitably connected with the so-called healthy lifestyle and, therefore, are points of interest of trainers and sportsmen, as much as amateur fitness enthusiasts. Instagram^®^, as an image-oriented social media platform, is especially predisposed to share and promote this form of content. An Australian study showed that they help promote physical activity awareness campaigns, enabling broader recipient reach and recruitment than traditional promotion methods [13]. A similar trend was observed in our study. Posts qualified as fitness related had the second-highest number of followers, likes, and engagement rates within the major categories of posts. At the same time, posts containing motivational quotes performed best but many related to a healthy lifestyle and staying fit. Therefore, they could be treated as part of a broader context—health promotion. However, not all studies are enthusiastic about the efficiency of health promotion with attempts to motivate with such content. For example, Tiggemann found that presenting body images on social media with inspirational intention decreased body satisfaction in undergraduate students [14]. Interestingly, among all performance measures, cooking recipes were the most commented ones, resulting from the need to clarify the recipes and answer their questions.

### Study Limitations

The main limitation of this study is the considerable heterogeneity of the results, especially seen in the number of followers between the account, ranging from few to a few hundred thousand. However, very dynamic changes in performance measures of each post occurring in real time when viewers react to the posted content impede data gathering. That is why we decided to search for the newest post, maintaining a proper time. It also reflects the real-life presence of the content.

The lack of a dedicated quality assessment tool is also an issue. At the same time, the image-based construction of the Instagram^®^ platform, in which captions serve as additional layers of information, impedes proper contextual assessment with the use of tools designed for text-based social media platforms.

## 5. Conclusions

The quality of Instagram^®^’s nutrition-related content is extremely low. Therefore, random post searches cannot provide the viewer with valuable nutrition information due to many distorting contents. 

With its potential to reach many viewers, there is a need to share more high-quality content through social media channels. More action toward the provision of proper-quality information should be undertaken by nutrition-oriented organizations and professionals to share it with social media users.

## Figures and Tables

**Figure 1 healthcare-10-00397-f001:**
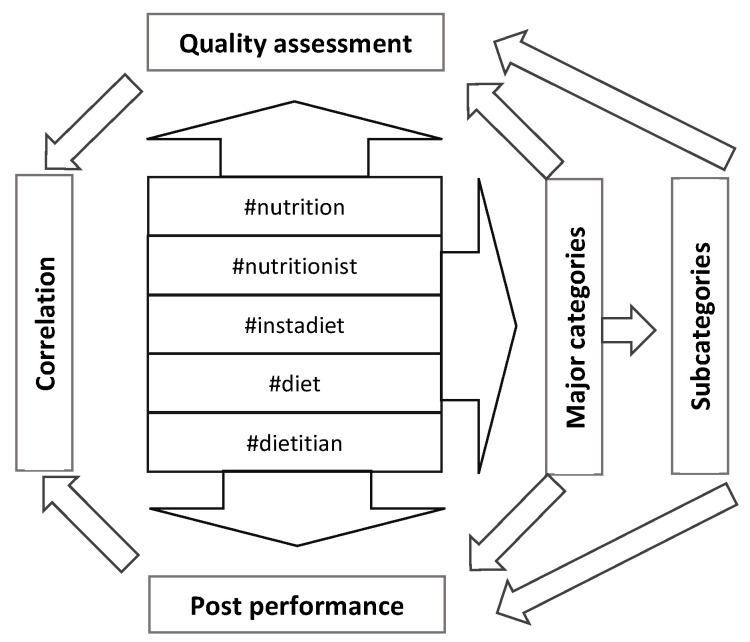
Study scheme.

**Table 1 healthcare-10-00397-t001:** Popularity measures of posts in hashtag groups.

Category and Subcategory Name	Whole Group	#Nutrition	#Nutritionist	#Instadiet	#Diet	#Dietitian
	Category					
Nutrition and dietetics	63.8%	47.0%	62.3%	79.8%	55.0%	71.7%
Fitness	8.2%	11.5%	6.3%	3.6%	15.1%	4.9%
Motivation	4.8%	6.4%	10.9%	1.2%	3.4%	2.5%
Other	23.2%	34.1%	20.5%	15.4%	26.5%	20.9%
	Subcategory					
Dietary advice	15.6%	14.9%	21.3%	14.2%	7.6%	20.1%
Coaching	3.0%	2.3%	1.3%	2.0%	6.7%	2.9%
Cooking recipe	10.0%	8.8%	18.8%	5.1%	4.6%	12.7%
Photo	42.8%	54.9%	21.8%	53.8%	51.3%	33.2%
Motivation	5.6%	7.0%	11.7%	2.0%	5.0%	2.5%
Fitness	0.3%	None	0.4%	0.8%	0.4%	None
Other	22.7%	12.1%	24.7%	22.1%	24.4%	28.6%

**Table 2 healthcare-10-00397-t002:** Spearman’s rank correlation.

	Quality	
Likes	0.128	*p* =< 0.001
Comments	0.096	*p* =< 0.001
Followers	0.056	*p* = 0.026
Like ratio	−0.030	*p* = 0.154
Comment ratio	0.006	*p* = 0.23
Engagement rate	0.02	*p* = 0.163

Like ratio = number of likes/number of followers; comment ratio = number of comments/number of followers; engagement rate = number of likes + number of comments/number of followers.

**Table 3 healthcare-10-00397-t003:** ANOVA analysis for performance measures by category.

	Nutrition and Dietetics	Fitness	Motivation	Other	
	*n* = 759	*n* = 97	*n* = 57	*n* = 276	
Likes	116	101	88	99	*p* = 0.91
Comments	5	4	6	5	*p* = 0.82
Followers	2728	3451	4355	3328	*p* = 0.63
Like ratio	13.3%	18.6%	30.6%	13.9%	*p* = 0.03
Comment ratio	5.8%	5.7%	4.5%	6.1%	*p* = 0.59
Engagement rate	14.0%	21.7%	32.7%	14.5%	*p* = 0.03

Like ratio = number of likes/number of followers; comment ratio = number of comments/number of followers; engagement rate = number of likes + number of comments/number of followers.

**Table 4 healthcare-10-00397-t004:** ANOVA analysis for performance measures by subcategory.

	Dietary Advice	Coaching	Cooking Recipe	Photo	Motivation	Fitness	Other	
	*n* = 186	*n* = 36	*n* = 119	*n* = 509	*n* = 66	*n* = 4	*n* = 269	
Likes	112	55	158	83	366	20	82	*p* =< 0.001
Comments	5	2	11	4	7	1	4	*p* =< 0.001
Followers	3144	2154	5004	2509	5478	684	2502	*p* = 0.68
Like ratio	12.7%	11.3%	11.9%	15.6%	27.4%	9.8%	12.9%	*p* =< 0.001
Comment ratio	6.0%	4.7%	7.8%	5.0%	4.6%	3.9%	6.9%	*p* = 0.003
Engagement rate	13.4%	11.7%	12.7%	16.8%	29.3%	10.5%	13.6%	*p* = 0.001

Like ratio = number of likes/number of followers; comment ratio = number of comments/number of followers’ engagement rate = number of likes + number of comments/number of followers.

## Data Availability

All data come from publicly available, open profiles on Instagram^®^ platform.

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
