# Peer review of "Can Social Media Profiles Be a Reliable Source of Information on Nutrition and Dietetics?"

_healthcare, 2022, doi:10.3390/healthcare10020397_

Round 1
Reviewer 1 Report
Review of the article: Can social media profiles be a reliable source of information on nutrition and dietetics
The idea to study is good.
Chapter - materials and methods
Make a more detailed characterization of the tested material:
Date of analysis.
What period do the data come from
Characterize the studied group, e.g. age, nationality.
Consider adding a research schema to make your work transparent.
Statistical analysis
Please explain the sentence “Basic measures - mean as well as median values were calculated, and presented as such, because of significant differences between them”. In statistics, the mean or median of the dependence on the normality of the distribution is used.
Chapter - results
Add tables with a results. Consider adding Table 1 from supplementary material
Supplementary material
Complete missing information in Table 2, Table 3 and Table 4. For example P value
Explain the abbreviations below the tables 2 and 4.
Editorial errors should be corrected.
Author Response
The idea to study is good.
We would like to thank the reviewer for appreciation of the study concept.
Chapter - materials and methods
Make a more detailed characterization of the tested material:
- Date of analysis. What period do the data come from
The data were gathered between 25-26th November 2018. The post were searched for and assessed over 24-hour period. We entered appropriate corrections to the text.
- Characterize the studied group, e.g. age, nationality.
It was impossible to gather data regarding the study population, which in this case would be the post authors’ data. The publicly available data on Instagram® regarding requested characteristics are not required by the app and therefore don’t have to be presented.
- Consider adding a research schema to make your work transparent.
A scheme explaining the research methodology has been added to the text
Statistical analysis
- Please explain the sentence “Basic measures - mean as well as median values were calculated, and presented as such, because of significant differences between them”. In statistics, the mean or median of the dependence on the normality of the distribution is used.
We would like to thank the reviewer for noting this important issue. We have reevaluated the normality of distribution and agree with the comment. We have decided to leave the mean values as the ones properly reflecting the distribution. We entered appropriate corrections to the text.
Chapter - results
- Add tables with a results. Consider adding Table 1 from supplementary material
We’ve added the table directly into the manuscript
Supplementary material
- Complete missing information in Table 2, Table 3 and Table 4. For example P value
Explain the abbreviations below the tables 2 and 4.
The tables have been completed with the missing information
Editorial errors should be corrected.
The manuscript has been rechecked for editorial and language mistakes
Reviewer 2 Report
Social media are indeed growing, worldwide platforms for unlimited exchange of various content. The authors target nutrition and dietetics. The manuscript is well written, but lacks two crucial aspects, including consent and abuse. The consent is key, because adolescents have access to this platform. Abuse is quite common in social media and preventing it is essential. This needs to be emphasized in the manuscript. The quality of Instagram data is poor and bias and overlaps are missing in the manuscript. Thus, a method to assess the credibility of the reports is key for social media use. The 1189 posts need to be analyzed using an AI supported screening to highlight credibility and overlaps.
Author Response
Social media are indeed growing, worldwide platforms for unlimited exchange of various content. The authors target nutrition and dietetics. The manuscript is well written, but lacks two crucial aspects, including consent and abuse. The consent is key, because adolescents have access to this platform. Abuse is quite common in social media and preventing it is essential. This needs to be emphasized in the manuscript.
We would like to thank the reviewer for appreciation of the text. We do agree that consent and abuse are key, therefore after primary search of the posts in each hashtag group they have been screened for unwanted content, especially for commercials, which has been stated in the text. Even though we did not find any sign of abusive or hazardous content, these would have been discarded and excluded from further assessment. We’ve entered appropriate corrections in the text. We also took into consideration only these posts which were open to public view, thus with implied consent.
The quality of Instagram data is poor and bias and overlaps are missing in the manuscript. Thus, a method to assess the credibility of the reports is key for social media use. The 1189 posts need to be analyzed using an AI supported screening to highlight credibility and overlaps.
We would like to thank the reviewer for this comment. We also recognize the lack of dedicated quality assessment tool as a limitation of this study, however Instagram® is based on images while captions serve as an additional layer of information. This impedes proper contextual assessment compared to other social media i.e Twitter®, Facebook® or other based on textual content. We’ve added a comment on this issue to the text.
Round 2
Reviewer 1 Report
The suggestions for the manuscript were taken into account. However, the manuscript still requires editorial correction, especially about punctuation errors. In Table 1, please complete the title of the first column.
Author Response
We would like to thank the reviewer for taking the time to re-review the manuscript.
We have decided to use an external language correction service to improve clarity, readability, and punctuation errors.
We have also completed the titles in the table
Reviewer 2 Report
The authors properly addressed the comments and suggestions of the reviewers.
Author Response
We would like to thank the reviewer for taking the time to re-review the manuscript.
We have decided to use an external language correction service to improve clarity, readability, and punctuation errors.